# Aerodynamic Study of Velopharyngeal Insufficiency in 22q11.2 Deletion Syndrome

**DOI:** 10.3390/jpm14060620

**Published:** 2024-06-10

**Authors:** Salvatore Allosso, Massimo Mesolella, Giovanni Motta, Giuseppe Quaremba, Rosaria Parrella, Martina Ricciardiello, Sergio Motta

**Affiliations:** 1Unit of Otorhinolaryngology, Department of Neuroscience, Reproductive Sciences and Dentistry, University Federico II of Naples, 80131 Naples, Italy; rosparrella@gmail.com (R.P.); martinaricciardiello@icloud.com (M.R.); 2Unit of Otorhinolaryngology, Department of Mental and Physical Health and Preventive Medicine, University Luigi Vanvitelli, 80131 Naples, Italy; giovannimotta95@yahoo.com; 3Department of Advanced Biomedical Sciences, Federico II University of Naples, 80131 Naples, Italy; quaremba@unina.it (G.Q.); sermotta@yahoo.com (S.M.)

**Keywords:** velopharyngeal insufficiency, 22q11.2 deletion syndrome, cleft palate, speech aerodynamics

## Abstract

Objectives: We aim to verify velopharyngeal sphincter function in 22q11.2 deletion syndrome patients (22q11.2DS) to establish correlations between aerodynamic and perceptual measures of nasality, and to identify aerodynamic measures differentiating typical from atypical velopharyngeal behavior. Methods: Eleven subjects with 22q11.2DS and twenty similar-age control subjects were recruited. The aerodynamic measures were mean Sound Pressure Level, air pressure peak, pressure wave duration, airflow pattern and nasal airflow during the sequence /pi/. The nasality perceptual measures were rhinolalia, rhinophony and nasal air escape. Results: Airflow patterns and perceptual measures were statistically different in the two groups. Pressure wave duration and air pressure peak were lower in study subjects than in controls. Air pressure peak and nasal airflow were negatively correlated with rhinolalia; pressure wave duration was negatively correlated with nasal air escape and rhinolalia in 22q11.2DS patients. Conclusions: This aerodynamic study identified velopharyngeal qualitative and quantitative dysfunctions, suggesting heterogeneous models of velopharyngeal function in syndromic subjects as compared to controls.

## 1. Introduction

Chromosome 22q11.2 deletion syndrome (22q11.2DS) includes an extremely variable spectrum of upper aero-digestive tract disorders [1,2,3], representing the most frequent genetic cause of velopharyngeal insufficiency (VPI) [4,5]. The 22q11DS patients can show speech disturbances due to VPI as open rhinolalia, nasal air escape and an increase in nasal resonance, known as hyper-rhinophony or open rhinophony. A broader consensus has recently been reached concerning the anatomical causes of VPI in 22q11DS, suggesting a predominance of palatal submucosal clefts in the classical and occult variants [6,7,8]. The effects on speech of VPI have been investigated for more than 50 years, and several contributions on speech aerodynamics in palatal cleft have been published, mostly as compared with control data [9,10,11,12,13,14]. Aerodynamic tests (pressure–flow) have demonstrated scientific validity in the evaluation of the closure of the palatine velum (VP) during speech, providing indirect indications about the VP air orifice (VPA) and the closure time of the VP [15,16].

The pressure–flow technique, introduced by Warren and Dubois [9], constitutes the benchmark for all studies dedicated to the aerodynamics of speech production in individuals with VPI. This technique can define the velopharyngeal port area, especially for relatively small velopharyngeal orifice sizes [10,11], and the respiratory functional adaptations with respect to VPI [12,15]. Aerodynamic investigations in 22q11.2DS individuals have not yet been carried out, nor has any obvious correlation been verified between aerodynamic and speech perceptual findings in this syndrome. The high frequency of palatal submucosal anomalies makes correct functional speech assessment difficult in 22q11DS patients, with potentially unfavorable consequences on indications as to the most suitable rehabilitation strategies [12,15,16].

In the present study, we set out to establish the functioning of the velopharyngeal sphincter and its aerodynamic behavior during speech articulation in patients with 22q11.2DS compared to non-syndromic controls. We also sought to identify any correlations between aerodynamic and speech perceptual measures in syndromic individuals and to verify which aerodynamic measures, if any, could allow us to assess velopharyngeal sphincter function and to distinguish typical from atypical velopharyngeal behavior.

## 2. Materials and Methods

The Institutional Review Board of Naples University School of Medicine approved this study, which was performed in accordance with the Helsinki Declaration of 1975. Children’s consent, together with written informed consent and permission by parents/guardians, was obtained for all subjects recruited in the study.

In a recent mono-centric investigation, published elsewhere [7], twenty-five 22q11.2DS individuals were enrolled, in 2018, to identify the functional effects of otolaryngological anomalies and their relations with infection susceptibility in this syndrome.

Patients under the age of 3 years, or those for whom informed consent had not been provided by their parents or guardians, were excluded.

A total of 11 of the 25 individuals (7 males and 4 females, mean age 12.4 ± 3.8 SD) completed the functional speech assessment with the aerodynamic analysis and were included in the present study (Table 1). A total of 20 healthy individuals (13 males and 7 females, mean age 12.2 ± 3.5 SD) were selected, according to the age/sex criteria of the case–control type. All enrolled individuals underwent perceptual speech production assessment by three expert speech pathologists.

Each study subject was invited to produce about 5 to 10 Italian poly-term sentences containing vowel sounds, and fricative, affricated and plosive consonants in a silent room. We used up to 10 bisyllabic/trisyllabic words for less collaborative individuals (7 cases, comprising 5 control subjects and 2 syndromic cases). The speech sample was recorded using the Voice Analysis module of the Daisy 3.6 program [13] (Biomedica Amplifon, Ripamonti, 133, 20141, Milano, Italy). Three experienced speech pathologists assessed and perceptually graded rhinolalia, rhinophony and nasal air escape, and repeated the same assessment after 15 days, according to the classification by Massari [17]. Each expert listened to the speech sample recordings of the 31 recruited individuals in a randomized order, for as long as necessary, for the purpose of perceptual evaluation. Intra-judge and inter-judge reliability was verified by calculating Cohen’s K. Overall, Cohen’s K was 0.88, after excluding one of the three experts, due to lower intra-evaluator reliability. The aerodynamic examination was carried out using the system “Aerophone II” (FJ Electronics, Vedbaek, Denmark), equipped with phonatory flow physical calibration, during the repeated production of the phoneme /pi/. The patient, placed in a sitting position in front of the equipment, was asked to inhale a small amount of air before performing the verbal articulatory test. Then, wearing the facial mask and having inserted the intra-oral tube, the patient was invited to produce the established verbal articulatory sequence (/p/). A facial mask connected to a transducer allowed the simultaneous recording of Sound Pressure Level (dB SPL), endo-oral pressure (cm of H_2_O) and phonatory airflow (milliliters per second).

All individuals’ tracings were supplied to one of the authors of the present study (S.M.) in blind conditions. A segment comprising at least five complete cycles of the phoneme /pi/ was used for the analysis. SPL was calculated at the level of vowel sound /i/, while air pressure peak and pressure wave duration were extracted and calculated from the endo-oral pressure tracing. Airflow morphologic pattern was classified as follows: (a) type 1, well-defined morphology with biphasic pattern and airflow peak following air pressure wave; (b) type 2, well-defined morphology with variations between contiguous cycles; (c) type 3, evident trace atypia and scarcely recognizable biphasic pattern; and (d) type 4, severe atypia with unidentifiable biphasic pattern. Any airflow during the implosive phase of /p/, indicating nasal airflow, was calculated. The airflow pattern classified as “1” was considered indicative of adequate velopharyngeal function. Data from perceptual assessment and aerodynamic examination were associated with the individual cases and grouped into the syndromic and non-syndromic control cases for subsequent statistical analysis.

Statistical analysis was performed using the software IBM SPSS Statistics, v.20,0 (IBM Corp., Armonk, NY, USA). A Mann–Whitney test was used to compare the quantitative aerodynamic variables of syndromic and control subjects. The homoscedasticity of variables was assessed by Levene’s test. A comparison of qualitative aerodynamic and perceptual categorical variables was carried out by a χ^2^ test. Spearman’s correlation coefficient was used to ascertain the existence of any correlation, and two-tailed significance was calculated. Discriminant analysis was carried out by entering all variables and selecting the best set of discriminating variables through a stepwise method. The discriminant scores were derived by maximizing the quadratic distance of Mahalanobis from the centroid of the two clusters. All *p*-values were two-sided and values less than 0.05 were considered significant.

## 3. Results

Ten out of the eleven study patients showed morphological and/or functional palatal anomalies (Table 1). Objective oro-pharyngoscopy and/or rhino-pharyngoscopy findings were suggestive of an occult submucosal cleft in 6/11 cases (54%). Descriptive data and statistical comparisons for all the studied variables are reported in Table 2 and Table 3: airflow patterns showed morphological anomalies in 7/11 (64%) syndromic individuals with a significant difference between the two groups (Table 2); air pressure peak and pressure wave duration were significantly higher in controls compared to syndromic subjects (Table 3). Speech perceptual anomalies significantly differentiated syndromic subjects from controls (Table 2).

The results of the correlation analysis are shown in detail in Table 4. Overall, air pressure peak and pressure wave duration showed a significant negative correlation with rhinolalia. Nasal airflow showed a positive correlation with airflow pattern, nasal air escape and rhinolalia. Pressure wave duration is negatively correlated with nasal air escape.

Discriminant analysis correctly classified 7/11 cases (63.6%) of the study group and 20/20 cases (100%) of the control group. The correctly classified syndromic cases showed atypical airflow patterns with discriminant scores between 0.929 and 4.009. A type 1 airflow pattern distinguished the other 24 cases studied, i.e., 20 non-syndromic and 4 syndromic cases. Two of the four syndromic subjects classified as controls had discriminant scores (−0.645 and −0.360) falling in the range of control subjects (from −0.117 to—2.180), with a pressure wave duration greater than 200 ms; the other two patients showed borderline scores (−0.011 and 0.200), with a pressure wave duration of less than 200 ms.

## 4. Discussion

In the present investigation, an aerodynamic and perceptual study of speech was carried out in 22q11.2DS individuals and controls. Ten out of the eleven study cases showed palatal anomalies. In six patients (54%), diagnosis of occult submucosal cleft was based on a clinical examination by oropharyngoscopy and in four out of the six cases, also by flexible nasopharyngoscopy. In this regard, it must be emphasized that more nuanced structural anomalies underlie occult submucous clefts compared to classic “open” submucosal forms with Calnan triad (uvula bifida, soft palate muscle diastasis, palpable notch in hard palate). Therefore, occult submucous cleft anomalies can be exclusively identifiable through intra-operative surgical dissection and, as such, can only be presumed (position and insertion anomalies of the levator palati and/or uvula muscle; minimal alterations to the palatal bone posterior border) in non-operated cases after conventional clinical and instrumental examination [6,16,17,18].

A predominance of “posterior” palate clefts, with a considerable portion of cases consisting of submucosal forms and transitional expressions between the open and occult variant, has already been demonstrated in 22q11.2DS individuals [19]. Imaging data also corroborated the hypothesis of velopharyngeal sphincter sub-clinical anomalies in 22q11.2DS patients [19,20]. Our data regarding the high frequency of palatal anomalies in 22q11.2DS individuals are largely consistent with those of other authors [1,2,20,21,22,23].

Statistical analysis excluded significant differences in SPL between the study group and controls, which would make the aerodynamic data difficult to compare. From an aerodynamic point of view, we found a statistically significant reduction in air pressure peak during the implosion of /p/ in 22q11.2DS subjects, as compared to controls. In this regard, Dalston and colleagues [12] and Laine et al. [15] showed that endoral pressure during the plosive phase of voiceless consonants was negatively influenced by the VPI degree. In both investigations, a pressure greater than 3 cm H_2_O was recorded in most of the study patients, as well as in cases with a grossly inadequate velopharyngeal competence; according to the authors, this value would constitute the minimum threshold for pressure-sensitive consonant production. Data related to pressure wave duration deserve further consideration. In the present study, the mean value of this measure was significantly higher in controls, compared to study subjects. Contrary to our findings, Warren et al. [11], using the carrier word “hamper”, observed an increase in the average pressure duration in all the cleft palate sub-groups examined compared to a normal control group. These findings were statistically significant only for the adequate and borderline palate–pharyngeal function sub-groups, but not for the sub-group with an obvious palate–pharyngeal inadequacy. Interestingly, the control group pressure duration means value (0.140 s) in the Warren and colleagues study [11] was much lower than that recorded in our investigation. Therefore, an interference in results obtained in the two studies, due to speech sample diversity and differences in investigation techniques, cannot be excluded. In the study by Warren et al. [11], a particular focus was placed on the temporal relationship between nasal airflow duration during the emission of the sound /m/ and the endoral pressure duration of the sound /p/. However, the attribution of nasal airflow to the emission of nasal (/m/) or oral bilabial sound (/p/) is not easy to define in the case of nasal air escape due to VPI, which occurs during the plosive phase of /p/.

Nasal airflow data obtained in the present study, as compared to other surveys focusing on the aerodynamics of VPI [10,13,14,15,16,23,24,25,26,27,28], require more in-depth consideration. In our investigation, this measure has been correlated with the central point of the pressure wave or its peak (Figure 1, Figure 2, Figure 3 and Figure 4). In our opinion, any nasal air loss can be easily identified and quantified during the implosive phase of a pressure-sensitive sound such as /p/; in only ten of the twenty control subjects, measurable but minimal (lower than 20 mL/s) airflow was detected during this phase. The detection of occasional episodes of nasal airflow during the production of /p/ in a syllabic context was noted in children without clinical VPI by Searl and Carpenter [24], with values even higher than 45 cm^3^/s, as also reported in adults [13]. In the syndromic patients recruited for the present study, nasal airflow showed a remarkable variability: in 8 out of the 11 study patients, nasal airflow was undetectable (7 cases) or lower than 20 mL/s (1 case), while three patients showed a value higher than 100 mL/s up to a maximum of about 1 liter/s (Figure 2). Warren [10] registered nasal airflow peaks of more than 175 cm^3^/s during the production of /p/ through the pressure–flow technique in study patients with VPI. According to this author, patients classified as having “adequate closure” produced /p/ with values below 155 cm^3^/s of airflow.

In the present study, airflow patterns statistically differed between controls and 22q11.2DS patients. The airflow trace showed a biphasic pattern in control subjects, with a first peak-like component corresponding to the explosive phase of /p/, followed by a plateau-like phase due to the vowel emission (Figure 1). Seven out of eleven study patients did not exhibit this typical biphasic pattern, lacking the presence of nasal airflow synchronous with the endo-oral pressure rise during the implosive phase of /p/ (Figure 2) or the attenuation of the peak-like component (Figure 3). In the present study, airflow tracing focused on speech-distorting effects, particularly due to the interaction between nasal airflow and endo-oral pressure in syndromic subjects. In our opinion, the exclusive quantification of velopharyngeal defects or the definition of threshold values between normal and pathological subjects, as in studies concerning one or more aerodynamic variables such as ref. [13], do not allow a general overview of the pathological relations between these physical forces. Morphological analysis can also be particularly useful in cases where the diagnosis of VPI is made difficult by the presence of occult clefts, frequently found in 22q11.2DS patients, in whom quantitative aerodynamic measures can fall within the norm.

Airflow recorded during the implosive phase of /p/ (nasal airflow) showed a highly positive correlation with anomalies in sound-intensity tracing, indicative of a nasal turbulence noise created by trans-nasal air escape. Turbulence noise, in turn, positively correlated with airflow pattern anomalies; morphological airflow variations therefore positively correlated with nasal airflow (Table 4).

The lack of correlation between pressure wave duration and air pressure peak observed in our investigation is rather surprising. Such data may be interpreted based on the extreme variability of velopharyngeal sphincter function in study patients: cases with relatively adequate pressure duration but with low peak pressure or, conversely, patients in whom a reduced pressure duration is associated with a relatively high air pressure peak. These findings confirm the development of heterogeneous compensation mechanisms in the recruited patients, as supported by other studies dedicated to VPI aerodynamics [12,16,25]. Among the perceptual measures, nasal air escape correlated positively with rhinolalia (Table 4). Otherwise, rhinophonia, whose increase clearly differentiated syndromic subjects from control subjects (Table 3), showed no significant correlation with other perceptual measures (Table 4). This lack of correlation is of considerable interest as it indicates a dissociation between nasality-related speech anomalies (rhinolalia and rhinophonia) in the study sample. Such a finding is probably linked to anatomical conditions found in 22q11.2DS patients (platybasia with a greater amplitude of nasopharyngeal lumen and/or a reduction in the thickness and/or length of the soft palate) that may lead to an increase in nasal resonance in the absence of speech distortions and/or nasal air escape [19,20,21,22,23,24]. Therefore, only two of the perceptual measures studied, rhinolalia and nasal air escape, showed a statistical correlation with aerodynamic variables, indicating a high correspondence between aerodynamic findings and articulatory nasal airflow-induced defects (Table 4).

Discriminant analysis identified a binomial (airflow pattern and air pressure wave duration), whose orthogonal combination allowed us to classify the 20 control subjects as such, while four of the 11 syndromic subjects were classified in the group of controls. In other words, the discriminant analysis made it possible to define an aerodynamic binomial capable of differentiating the subjects with typical velopharyngeal function (20 control cases and two syndromic subjects) from those with evident functional anomalies (7 syndromic cases) or with borderline velopharyngeal behavior (2 syndromic cases). In the present investigation, airflow pattern and pressure wave duration differentiated dysfunctional profiles of the velopharyngeal sphincter from identified borderline cases (Figure 4).

The 22q11.2DS represents a considerable challenge for clinicians and researchers who investigate the effects of palatal anomalies on speech production, contemplating cases of extremely variable severity, up to borderline expressions of velopharyngeal dysfunction. Palatal anomalies found in the study patients mainly concerned the posterior palate, with a prevalence of occult submucosal clefts. The use of a facial mask was not well accepted by many of the 22q11.2DS individuals initially recruited, and this led to the reduction of the study sample. Therefore, due to the limited number of study subjects, stratification based on the type of anatomical defect was not affected. In all the study patients, speech therapy was carried out or was still in progress; moreover, four of the eleven recruited subjects had previously undergone surgery on the primary palatal defect. It was therefore not possible to establish how much velopharyngeal sphincter function was influenced by speech therapy and/or by surgical treatment in the sample studied.

Deviating aerodynamic measures can represent reliable VPI indicators, which are of even greater relevance in cases characterized by occult palatal defects, as often occurs in 22q11.2DS. Aerodynamic findings clearly suggest not only the extent of sphincter insufficiency but can also help the clinician and speech pathologist in evaluating the effectiveness and limits of the rehabilitation treatment performed in such cases. Data obtained from the present investigation should, of course, be considered preliminary, since they concern a small number of subjects affected by a rare syndrome.

In this regard, it should be noted that not all patients affected by the syndrome were able or managed to collaborate for the purpose of carrying out the aerodynamic examination. For this reason, children under the age of 3 and those who did not demonstrate sufficient collaboration skills were excluded from the study.

With these limitations, the sample we studied is nevertheless representative of the different levels of severity of palatal anomalies that the syndrome can involve.

Further studies on larger samples are required to verify the reliability of the aerodynamic data obtained in this study and to establish any relationships between morphological and aerodynamic findings.

Our data suggest the possibility of applying the aerodynamic study in all cases of patients suffering from 22q11.2DS syndrome, to define and quantify any functional deficits, even minimal ones, which may escape psychoacoustic examination. In this regard, an aerodynamic study, conducted alongside the speech therapy rehabilitation process, could define more precisely which cases may be candidates for optional pharyngoplasty surgical treatments, or the cases in which speech therapy has achieved optimal functional outcomes.

## 5. Conclusions

Velopharyngeal dysfunction in 22q11.2DS is often related to anatomical anomalies, which are clinically difficult to identify and can lead to highly variable dysfunctional effects. The aerodynamic data obtained in the present study showed uniform and highly repetitive speech sequences in control subjects. By contrast, heterogeneous velopharyngeal sphincter function was found in syndromic patients, with a frequently atypical airflow pattern. A reduced velopharyngeal sphincter performance, in terms of air pressure peak and pressure wave duration, mainly characterized the syndromic cases. The perception of rhinolalia and nasal air escape showed a well-defined correlation with some of the aerodynamic variables studied, unlike for rhinophonia. Finally, airflow morphological features and pressure wave duration differentiated subjects with typical velopharyngeal function from dysfunctional or borderline cases.

## Figures and Tables

**Figure 1 jpm-14-00620-f001:**
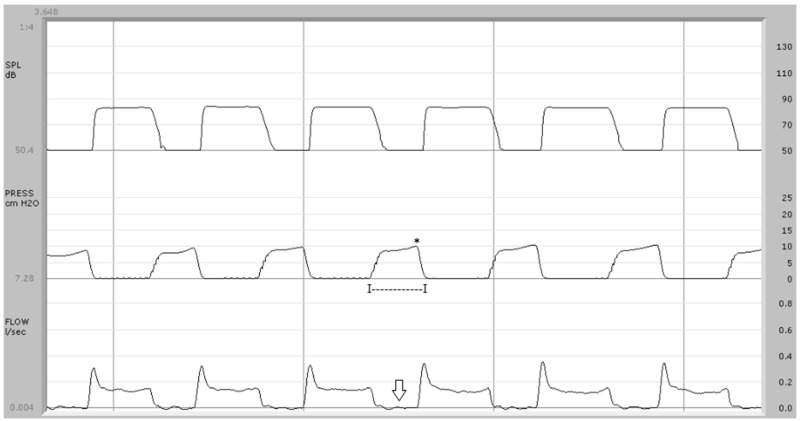
Aerodynamic exam in control subject: upper trace = sound pressure; center trace = endo-oral air pressure; bottom trace = airflow. The airflow trace (type 1) shows a biphasic pattern with a peak-like component corresponding to the explosive phase of the sound /p/. The pressure wave average duration is 270 ms (I—I) and the average pressure peak is 9.9 cm H_2_O (*). The average airflow during the implosive phase of /p/ is 0.002 L/s (arrow).

**Figure 2 jpm-14-00620-f002:**
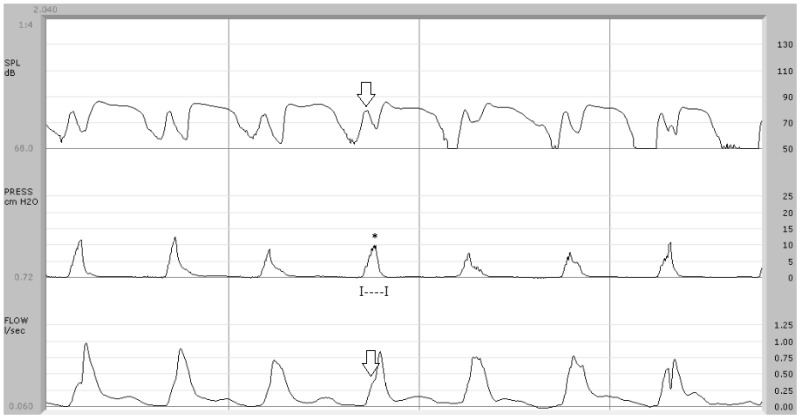
Aerodynamic findings in syndromic patient (case no. 1). In the upper trace, a component due to nasal airflow with a peak equal to 78 dB SPL is detectable (arrow). In the center trace, the air pressure wave shows a monophasic pattern with an average peak of 8.5 cm H_2_O (*) and a duration of 105 ms (I—I). Below, the airflow trace shows anomalies that make the typical biphasic pattern scarcely recognizable and an airflow peak during the implosive phase of /p/ with an average value of 0.388 L/s (arrow).

**Figure 3 jpm-14-00620-f003:**
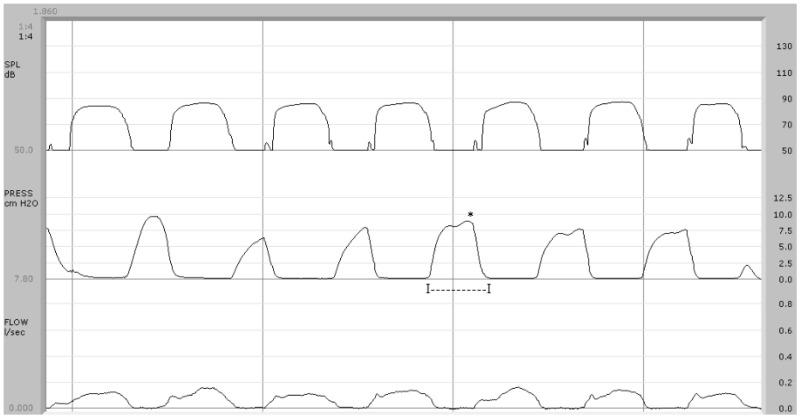
Syndromic patient (case no. 4). Below, the airflow trace is “beheaded” with a not clearly recognizable phase corresponding to the explosion of /p/. There are no findings indicative of nasal airflow during the implosive phase of /p/. The pressure wave duration is, on average, 240 ms (I—I) and the pressure peak is 8.1 cm H_2_O (*).

**Figure 4 jpm-14-00620-f004:**
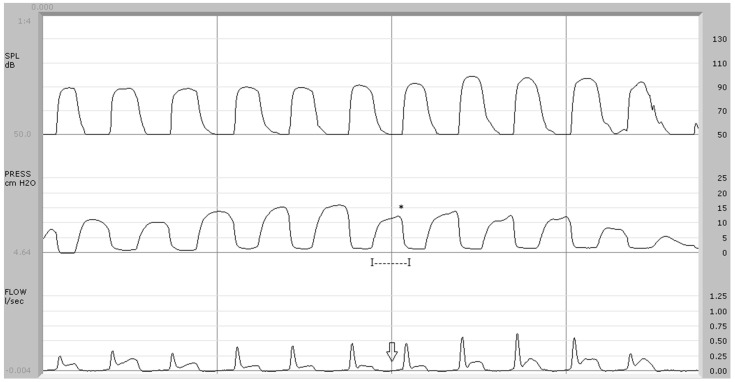
Syndromic patient (case no. 8). In the center trace, the average pressure peak (*) is 14.2 cm H_2_O and the pressure duration is 194 msec (I—I). Below, the arrow indicates the absence of airflow during the implosive phase of /p/. Discriminant analysis classified this subject as “borderline” (discriminant score: −0.011).

**Table 1 jpm-14-00620-t001:** Anthropometric and clinical data of 22q11.2 DS individuals enrolled in the study.

Case No.	Gender	Age at Assessment	Palatal Anomaly
1	f	15	SCP
2	f	15	SCP
3	m	13	SS
4	m	5	USS
5	m	11	OCP
6	m	10	No anomaly
7	m	13	OCP
8	m	13	OCP
9	m	19	OCP
10	f	14	OCP
11	f	8	OCP

SCP = open submucous cleft palate; SS = staphyloschisis; USS = urano-staphyloschisis; OCP = occult submucous cleft palate.

**Table 2 jpm-14-00620-t002:** Aerodynamic variables in 22q11.2 DS individuals and controls.

Parameters	22q11.2 DS Individuals Median (IQR)	Controls Median (IQR)	P	Levene
SPL (dB SPL)	80 (12.0)	77.50 (5.8)	0.887	0.145
Air pressure peak (cm H_2_O)	8.10 (9.1)	11.65 (8.0)	0.049	0.664
Pressure wave duration (ms)	237.0 (63.0)	273.5 (91.5)	0.007	0.779
Nasal airflow (mL/s)	~0.0	~0.0	0.855	0.001

IQR = inter-quartile range; P = statistical significance with Mann–Whitney U-test; Levene = result of Levene’s test.

**Table 3 jpm-14-00620-t003:** Contingency tables for airflow pattern and perceptual variables in 22q11.2 DS individuals and controls.

Parameters	Class	Study Subjects (%)	Controls (%)	P
Airflow pattern	1	36.4	100	0.001
	2	18.2	0	
	3	36.4	0	
	4	9.1	0	
Rhinolalia	1	45.5	100	0.001
	2	36.4	0	
	3	18.2	0	
Rhinophony	1	18.2	100	<0.0001
	2	63.6	0	
	3	18.2	0	
Nasal air escape	1	63.6	100	0.015
	2	18.2	0	
	3	18.2	0	

P = statistical significance with χ^2^ test. Nos. 1 to 4 and 1 to 3, respectively, for airflow pattern and perceptual variables (rhinolalia, rhinophony, nasal air escape) represent the categories (see the methods section) assigned to the syndromic and control subjects.

**Table 4 jpm-14-00620-t004:** Analysis of correlations according to Spearman’s correlation coefficient.

		SPL	Peak Air Pressure	Pressure Wave Duration	Nasal Airflow	Airflow Pattern	Rhinolalia	Rhinophony	Nasal Air Escape
SPL	Coeff.	1.000	0.416	−0.265	−0.085	−0.283	−0.163	0.263	−0.023
	Sign.		0.204	0.204	0.805	0.399	0.633	0.434	0.948
Peak air Pressure	Coeff.		1.000	0.382	−0.295	−0.320	−0.691	0.095	−0.568
	Sign.			0.247	0.379	0.337	0.018	0.780	0.068
Pressure wave duration	Coeff.			1.000	−0.421	0.119	−0.618	−0.381	−0.658
	Sign.				0.197	0.726	0.043	0.247	0.028
Nasal airflow	Coeff.				1.000	0.637	0.613	0.331	0.704
	Sign.					0.035	0.045	0.320	0.016
Airflow pattern	Coeff.					1.000	0.487	0.000	0.527
	Sign.						0.128	1.000	0.096
Rhinolalia	Coeff.						1.000	0.154	0.895
	Sign.							0.651	0.000
Rhinophony	Coeff.							1.000	0.105
	Sign.								0.760
Nasal air escape	Coeff.								1.000
	Sign.								

## Data Availability

The original contributions presented in the study are included in the article, further inquiries can be directed to the corresponding author.

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
