# Peer review of "Aerodynamic Study of Velopharyngeal Insufficiency in 22q11.2 Deletion Syndrome"

_jpm, 2024, doi:10.3390/jpm14060620_

Round 1

Reviewer 1 Report

Comments and Suggestions for Authors

Thank you for the opportunity to review your manuscript. Your study contributes important findings to the understanding of speech production challenges in individuals with 22q11 deletion. I would recommend a few minor enhancements.

(1) Please expand the methodology section to include more detailed descriptions of the aerodynamic measurement techniques and the criteria for selecting participants

(2) Consider enhancing the discussion by comparing your findings with existing literature on non-syndromic cases of velopharyngeal insufficiency

(3) Please address potential biases or variables that could influence the outcomes of your study, such as the age of participants and the severity of their condition. Discussing these factors would strengthen the validity of your conclusions

(4) Consider the clinical implications of your findings. Discuss how the novel insights into aerodynamic measures could influence treatment planning or outcomes for these patients

Author Response

Thank you for the opportunity to review your manuscript. Your study contributes important findings to the understanding of speech production challenges in individuals with 22q11 deletion. I would recommend a few minor enhancements.

  • Please expand the methodology section to include more detailed descriptions of the aerodynamic measurement techniques and the criteria for selecting participants.

I have included a more detailed explanation of how the test was carried out and the exclusion criteria for the study in the methods section. Thanks for the comment and suggestions.

  • Consider enhancing the discussion by comparing your findings with existing literature on non-syndromic cases of velopharyngeal insufficiency

Dear reviewer, in the literature there are no articles that clearly distinguish syndromic cases from non-syndromic ones regarding patients with velopalatal insufficiency. For this reason, a scientific comparison that allows us to identify the differences is not possible.

  • Please address potential biases or variables that could influence the outcomes of your study, such as the age of participants and the severity of their condition. Discussing these factors would strengthen the validity of your conclusions

Dear reviewer, in the discussion section (line 297) I discussed in detail the discussion regarding the limits relating to age and the differences in palatal insufficiency for the individual patients studied.

  • Consider the clinical implications of your findings. Discuss how the novel insights into aerodynamic measures could influence treatment planning or outcomes for these patients

Dear reviewer, I have included in the discussion (line 306) a comment regarding the future implications of the study as suggested. Thank you for the attention and time you have dedicated to our work.

Reviewer 2 Report

Comments and Suggestions for Authors

The authors present a case-control study, in order to describe the aerodynamic characteristics of velopharyngeal insufficiency of patients with 22q11.2 deletion syndrome, in comparison to healthy individuals. It is an interesting topic, that could add to the literature. However, there is a number of issues that need to be addressed, prior to consideration for publication.

  • The abstract is well structured. In the following sentence there is no verb, please correct it: “Air pressure peak and nasal airflow 23 negatively correlated with rhinolalia and pressure wave duration negatively correlated with nasal 24 air escape and rhinolalia in 22q11.2DS patients”.
  • The introduction provides a proper background, but it could probably be upgraded with more recent references. In the first two sentences of first paragraph, the references should be superscripted. Indeed, the literature regarding aerodynamics and speech in patients with this rare syndrome is rather poor. I could encounter only the following study, that probably should be referenced: “Cummings C, McCauley R, Baylis A. The Effect of Loudness Variation on Velopharyngeal Function in Children with 22q11.2 Deletion Syndrome: A Pilot Study. Folia Phoniatr Logop. 2015;67(2):76-82. doi: 10.1159/000438670”.

In the last sentence of second paragraph, a reference should be mentioned. 

  • Concerning the methodology part, please provide some information regarding the study period. Also, please, justify why you chose these specific 11 patients among the 25 that were included in the study you reference. The inclusion and exclusion criteria of study population should be clear.

In the first sentence of second paragraph of methods part, use number instead of “twenty-five”.

  • The results are clearly presented in the Tables 1-4.
  • In the discussion part the authors try to support their assumptions with notable references but it should be upgraded with some more recent articles that are encountered in the literature.  

We look forward to your revisions.

Comments on the Quality of English Language

The quality of English language is quite good. A review by a native English speaking editor/service check for a number of minor but significant grammatical/syntax issues would be appreciated.

Author Response

Dear reviewer, Thank you for the attention shown in reading the text and in the suggestions given.

The authors present a case-control study, in order to describe the aerodynamic characteristics of velopharyngeal insufficiency of patients with 22q11.2 deletion syndrome, in comparison to healthy individuals. It is an interesting topic, that could add to the literature. However, there is a number of issues that need to be addressed, prior to consideration for publication.

  • The abstract is well structured. In the following sentence there is no verb, please correct it: “Air pressure peak and nasal airflow 23 negatively correlated with rhinolalia and pressure wave duration negatively correlated with nasal 24 air escape and rhinolalia in 22q11.2DS patients”.
    • I modified the period by accepting your suggestion.
  • The introduction provides a proper background, but it could probably be upgraded with more recent references. In the first two sentences of first paragraph, the references should be superscripted. Indeed, the literature regarding aerodynamics and speech in patients with this rare syndrome is rather poor. I could encounter only the following study, that probably should be referenced: “Cummings C, McCauley R, Baylis A. The Effect of Loudness Variation on Velopharyngeal Function in Children with 22q11.2 Deletion Syndrome: A Pilot Study. Folia Phoniatr Logop. 2015;67(2):76-82. doi: 10.1159/000438670”.

In the last sentence of second paragraph, a reference should be mentioned. 

Dear reviewer, I have corrected the quotes not reported correctly, adding them where the reference was missing. I have also mentioned adding some ideas from the work suggested to me.

  • Concerning the methodology part, please provide some information regarding the study period. Also, please, justify why you chose these specific 11 patients among the 25 that were included in the study you reference. The inclusion and exclusion criteria of study population should be clear.
    • I have added: Patients under the age of 3 years or those for whom informed consent had not been signed by their parents or guardians were excluded.

In the first sentence of second paragraph of methods part, use number instead of “twenty-five”.

Correct.

  • The results are clearly presented in the Tables 1-4.
    • Thanks for your comment.
  • In the discussion part the authors try tosupport their assumptions with notable references but it should be upgraded with some more recent articles that are encountered in the literature.  
    • Dear reviewer, I performed a further literature review given the paucity of relevant articles. However, I have added two valid references (refs 21 and 25) which confirm and strengthen what was said in the discussion. Thanks for the suggestion.

We look forward to your revisions.

I corrected grammatical errors.

Round 2

Reviewer 1 Report

Comments and Suggestions for Authors

Thank you for submitting the revised manuscript. The revisions have significantly improved the clarity and robustness of your study. Your work provides valuable insights into the aerodynamic and perceptual measures of nasality in patients with 22q11, highlighting the differences in velopharyngeal functioning compared to controls. I find the manuscript well-written, and I believe it is ready for publication. Your contributions to this important area of study are valued.

Reviewer 2 Report

Comments and Suggestions for Authors

thank you for your revision

Comments on the Quality of English Language

The quality of English language is quite good. A review by a native English speaking editor/service check for a number of minor but significant grammatical/syntax issues would be appreciated.